# Double extreme-cum-median ranked set sampling

**Muhammad Zubair[1], Seyab Yasin[1], Afrah Al-Bossly[2], Asad Ali[3], Fathia Moh. Al Samman[4], Mohammed M. A. Almazah[5], Kanwal Iqbal[6]***

1 Department of Economics and Statistics, Dr Hasan Murad School of Management (HSM), University of Management and Technology, Lahore, Pakistan, 2 Mathematics, College of Science and Humanities in Al-Kari, Prince Sattam Bin Abdulaziz University, Al-Kharj, Saudi Arabia, 3 Department of Statistics, Govt Graduate College Abdullahpur, Faisalabad, Pakistan, 4 Department of Mathematics, College of Sciences, Northern Border University, Arar, Saudi Arabia, 5 Department of Mathematics, College of Sciences and Arts (Muhyil), King Khalid University, Muhyil, Saudi Arabia, 6 Department of Mathematics and Statistics, University of Lahore, Sargodha, Pakistan

* kanwaliqbal3110@gmail.com, kanwal.iqbal@math.uol.edu.pk

**Data Availability Statement:** The data underlying the results presented in the study are available in the given link https://mics.unicef.org/surveys.

**Funding:** Their authors extend Their appreciation to the Deanship of Scientific Research at King Khalid University for funding this work through Large

## Abstract

Extreme-cum-median ranked set sampling has been developed to address the problem of heterogeneity and outliers / extreme values. Double ranked set sampling has been suggested to obtain more reliable samples using the concept of degree of distinguishability. Dealing with heterogeneous and non-normal populations seems to be an area with a dearth of research. This article endeavors to address this research gap by introducing a new, improved ranked set sampling procedure that combines the aforementioned approaches, which is called double extreme-cum-median ranked set sampling. A simulation study for some symmetric and asymmetric probability distributions has been conducted. The results show that the newly proposed scheme performs better than its competitors under perfect and imperfect ranking, but the best performance has been observed for Weibull distribution with perfect ranking. An empirical study utilizing real-life data following skewed distribution was carried out. The real-life data results align well with the Monte Carlo simulation outcomes. Due to its flexible ranking options, the newly proposed technique is suggested for heterogeneous and non-normal populations.

## 1. Introduction

For estimating average pasture yields, McIntyre [1] developed ranked set sampling (RSS), whereas Takahasi and Wakimoto [2] provided a mathematical framework of RSS under perfect ranking. They also proved that the RSS estimator for the population mean is unbiased and more efficient than a simple random sampling (SRS) estimator, considering the same sample size, regardless of errors in ranking (imperfect ranking). Under imperfect ranking, Dell and Clutter [3] and David and Levine [4] reported same findings about efficiency of RSS. When ranking on research variables is exclusive, Stokes [5] suggested an auxiliary variable in RSS, resulting in lower study costs.

Research Groups Project under grant number
(RGP.2/41/45) and this study is supported via
funding from Prince Sattam bin Abdulaziz
University project number (PSAU/2023/R/1444).

Samawi et al. [6] proposed extreme ranked set sampling (ERSS) considering the situation when visual detection of *ith* ranked unit was difficult and expensive for sets consisting of a higher number of units, except the smallest and largest unit. Muttlak [7] introduced the median ranked set sampling (MRSS) to obtain a well-representative sample in the presence of heterogeneity. Al-Saleh and Al-Kadiri [8] presented a double ranked set sampling scheme (DMRSS) and demonstrated that ranking in the second stage is simpler than in the first stage of utilizing the idea of "degree of distinguishability" among sample observations. Ahmed and Shabbir [9] proposed extreme cum median ranked set sampling (EMRSS) to attain the benefits of both ERSS and MRSS.

## 2. Mathematical framework

### 2.1. Ranked Set Sampling (RSS)

To select a ranked set sample with a set size of *m*, initially, *m* simple random samples, each of size *m*, are chosen from the target population. After that, each sample is arranged in ascending order. The ranking is done without actually measuring the sample unit, instead employing a low-cost method like visual inspection or personal opinion. The *ith* sample's sample unit having judgment rank *i* ($i = 1, 2, \ldots, m$) is chosen for actual measurement and will be labeled as $Y_{i(i)}$. To get a ranked set sample of size $n = rm$, the entire process can be repeated *r* times (cycles) and final selected unit will be represented as $Y_{i(i)j}$ with $j = 1, 2, \ldots, r$. It is important to note that the set size *m* should be maintained low to expedite necessary informal ranks; for further information, see Mahdizadeh and Zamanzade [10]. The mean estimate for RSS is,

$$\widehat{\mu}_{y,RSS} = \frac{1}{mr} \sum_{j=1}^{r} \sum_{i=1}^{m} Y_{i(i)j}, \tag{1}$$

where $\widehat{\mu}_{y,RSS}$ is an unbiased estimator of $\mu_y$ and its variance with $\varphi_{(i)}^2 = \left( \mu_{y(i)} - \mu_y \right)^2$ is

$$var(\widehat{\mu}_{y,RSS}) = \frac{1}{mr} \left[ \sigma^2 - \frac{1}{m} \sum_{i=1}^{m} \varphi_{(i)}^2 \right], \tag{2}$$

where $Y_{i(i)j}$ means $i^{th}$ value of $i^{th}$ set from $j^{th}$ cycle and $\mu_{y(i)}$ represents mean of $i^{th}$ set for variable *Y*. In order to obtain more suitable and effective methodologies, RSS is modified to new structures. Hossain and Muttlak [11] proposed paired RSS to obtain more efficient estimates of population parameters when compared with SRS and RSS. Al-Odat and Al-Saleh [12] used some variations of RSS by using maximum and minimum values of data sets for estimating population parameters. Al-Saleh and Al-Omari [13] introduced multistage RSS for estimating the average yield of olives in West Jordan. Al-Nasser and Mustafa [14] proposed robust extreme ranked set sampling and proved its superiority over SRS, RSS and ERSS. Haq et al. [15] proposed mixed RSS for estimating the population mean. Sevinc et al. [16] introduced the concept of partial groups RSS to provide a more flexible sampling plan with independence of the set and sample sizes. Sohail et al. [17] proposed a class of ratio-type estimators of population mean for imputing the missing values under RSS. Al-Omari and Haq [18] suggested double L-RSS for more efficient estimation of population parameters when compared with SRS, RSS and L-RSS. Arnab et al. [19] introduced the concept of stratified two phase RSS. Noor-ul-Amin et al. [20] used even order ranked set sampling for the estimation of population parameter (mean). Iqbal et al. [21] proposed mixture regression cum ratio estimators of population mean under stratified random sampling.

Zamanzade and Mahdizadeh [22] used the RSS scheme with extreme ranks to estimate the population mean. Khan et al. [23] suggested mixture ranked set sampling for efficient estimation of population parameters (mean and median). Yang et al. [24] used the idea of RSS to

estimate log-extended exponential-geometric distribution parameters. Ali et al. [25] proposed a generalized family of estimators under classical ranked set sampling for the estimation of population mean. Ali et al. [26] proposed stratified extreme-cum-median ranked set sampling. Ali et al. [27] suggested generalized chain regression cum chain ratio estimators for population mean under stratified extreme-cum-median ranked set sampling.

Haq et al. [28] introduced paired double ranked set sampling for estimating population parameters. Noor-ul-Amin and Tayyab [29] suggested quartile paired double ranked set sampling for estimating population parameters for symmetrical and non-symmetrical distributions. Noor-ul-Amin et al. [30] introduced a modified extreme ranked set sampling scheme for more efficient estimates of the population mean. Further, the above-mentioned RSS strategies were also used to improve the efficiency of the control charts; one may see Riaz et al. [31], designed EWMA chart and investigated under various ranked set sampling strategies, was found to offer better detection ability, particularly with EWMA − 3[ERSS] and EWMA − 3 [DMRSS] showing greater efficiency. Abbas et al. [32] proposed new linear profiling methods under a Neoteric RSS (NRSS) scheme, with simulations showing that the new classical and Bayesian charts had better detection abilities, particularly the Bayesian charts under both perfect and imperfect NRSS. Mahmood et al. [33] investigated Phase I monitoring of linear profile parameters under various ranked set sampling approaches, showing that the proposed methods offered superior detection abilities compared to existing schemes, with a practical application demonstrated in a PV system. Touqeer et al. [34] aimed to enhance existing Phase I profile methods by using ranked set sampling strategies, showing improved detection of assignable causes, with a real-life application demonstrated using data from a grid-connected photovoltaic system. Mahmood et al. [35] proposed a Shewhart-type CV control chart under neoteric ranked set sampling, which demonstrated better detection ability compared to existing charts, with a real-life example illustrating its effectiveness.

## 2.2. Double Ranked Set Sampling (DRSS)

The following DRSS procedure was introduced by Al-Saleh and Al-Kadiri [8]. Assume that the population of interest, $Y$, has a size of $N$. Choose $m^3$ units and split each one into $m$ sets with size $m^2$. The units in each set are ranked from lowest to highest. Each set is subjected to the RSS technique to produce $m$ sets, each of size $m$ and selected units will be labeled as $Y_{i(i:m)}$. Once more, order the units of each set in ascending order. Each set is subjected to the RSS technique once more to produce $m$ units and selected units will be labeled as $Y_{i(i:m)}^{(i)(i:m)}$. One cycle has been completed. To get a sample with size $n = rm$, repeat the above steps $r$ times and selected units will be labeled as $Y_{i(i:m)j}^{(i)(i:m)}$. The sample mean under DRSS is given as,

$$\widehat{\mu}_{y,DRSS} = \frac{\sum_{j=1}^{r} \sum_{i=1}^{m} Y_{i(i:m)j}^{(i)(i:m)}}{rm}.$$ (3)

$\widehat{\mu}_{y,DRSS}$ is unbiased when the underlying distribution is symmetric about mean, and its variance is given as,

$$var(\widehat{\mu}_{y,DRSS}) = \frac{1}{rm}\left[\sigma_{(y)}^2 - \frac{1}{m}\sum_{i=1}^{m} \varphi_{(i:m)}^{2(i:m)}\right],$$ (4)

or

$$var(\widehat{\mu}_{y,DRSS}) = var(\widehat{\mu}_{y,RSS}) - \frac{1}{nm}\left[\sum_{i\neq 1}^{m} \sigma_{y(i,l:m)}^{(i,l:m)}\right],$$ (5)

where $\varphi_{(i:m)}^{2(i:m)} = \left(\mu_{y(i:m)}^{(i:m)} - \mu_{(y)}\right)^2$ and $\sigma_{y(i,l:m)}^{(i,l:m)} \geq 0$ for $i \neq l = 1,2,\ldots,m$, represents covariance between $Y_{i(i:m)j}^{(i)(i:m)}$ and $Y_{l(l:m)j}^{(l)(l:m)}$. For further details, see Haq et al. [27].

### 2.3. Extreme-cum-Median Ranked Set Sampling (EMRSS)

Ahmed and Shabbir [9] proposed the EMRSS approach to estimate the population mean. Assume that the population of interest, $Y$, has a size of $N$. Choose $2m$ separate random sets, each with a size of $2m$. In each set, sort the units in ascending order for the variable Y. To implement the ERSS method when $m$ is even, select the lowest ranked unit from each of the first $m/2$ sets and then choose the highest ranked unit from each of the subsequent $m/2$ sets. For the EMRSS procedure's completion, pick the $m^{th}$ unit from each of the $(m+1)$ to $(3m/2)^{th}$ sets, and select the $(m+1)^{th}$ unit from each of the last $m/2$ sets.

Choose the lowest rank unit from each of the first $(m-1)/2$ sets when $m$ is odd, the highest rank unit from each of the following $(m-1)/2$ sets, and the $m^{th}$ unit from $m^{th}$ set, which ends the ERSS process. To finalize the EMRSS procedure, choose the $(m+1)^{th}$ unit from the $(m+1)^{th}$ set, the $m^{th}$ unit from each of the $(m+2)^{th}$ to $\{((m+1)/2)+m\}^{th}$ sets, and the $(m+1)^{th}$ unit from each of the last $(m-1)/2$ sets. One cycle in this way is completed. To get a sample with size $n = rm$, repeat the procedure $r$ times. For even $(E)$ and odd $(O)$ sample sizes, the sample means using the EMRSS are given as,

$$\widehat{\mu}_{y,EMRSS(E)} = \frac{1}{2mr}\sum\nolimits_{j=1}^{r}\left[\begin{array}{c}\left(\sum_{i=1}^{\frac{m}{2}} Y_{i(1)j} + \sum_{i=\frac{m}{2}+1}^{m} Y_{i(2m)j}\right) + \\ \left(\sum_{i=m+1}^{\frac{3m}{2}} Y_{i(m)j} + \sum_{i=\frac{3m}{2}+1}^{2m} Y_{i(m+1)j}\right)\end{array}\right], \tag{6}$$

and

$$\widehat{\mu}_{y,EMRSS(O)} = \frac{1}{2mr}\sum\nolimits_{j=1}^{r}\left[\left(\sum_{i=1}^{\frac{(m-1)}{2}} Y_{i(1)j} + \sum_{i=\left\{\frac{(m-1)}{2}\right\}+1}^{m-1} Y_{i(2m)j} + Y_{m(m)j}\right)\right.$$
$$\left. + \left(Y_{m+1(m+1)j} + \sum_{i=m+2}^{\frac{(3m+1)}{2}} Y_{i(m)j} + \sum_{i=\frac{3(m+1)}{2}}^{2m} Y_{i(m+1)j}\right)\right]. \tag{7}$$

Both the estimators $\widehat{\mu}_{y,EMRSS(E)}$ and $\widehat{\mu}_{y,EMRSS(O)}$ are unbiased if the underlying distribution is symmetric about mean and the variances are given as,

$$var(\widehat{\mu}_{y,EMRSS(E)}) = \frac{\sigma_y^2}{2rm} - \frac{1}{4rm^2}\left\{\frac{m}{2}\left(\varphi_{(1)}^2 + \varphi_{(2m)}^2\right) + \frac{m}{2}\left(\varphi_{(m)}^2 + \varphi_{(m+1)}^2\right)\right\} \tag{8}$$

and

$$var(\widehat{\mu}_{y,EMRSS(O)}) = \frac{\sigma_y^2}{2rm} - \frac{1}{4rm^2}\left\{\frac{m-1}{2}\left(\varphi_{(1)}^2 + \varphi_{(2m)}^2\right) + \frac{m+1}{2}\left(\varphi_{(m)}^2 + \varphi_{(m+1)}^2\right)\right\}. \tag{9}$$

Based on the inspiration of Al-Saleh and Al-Kadiri [8] and Ahmed and Shabbir [9] studies, this study is intended to propose an extended version of EMRSS named double extreme-cum-

median ranked set sampling (DEMRSS), which is the combination of DRSS and EMRSS methods. The DEMRSS scheme is designed to provide a more representative sample and an efficient estimate of the population mean. It also aims to be a more competitive sampling strategy than existing methodologies. Moreover, this methodology is more appealing in situations where the population is heterogeneous and contains outliers.

## 2.4. Proposed sampling scheme

The framework of the DEMRSS is described as follows: assume that the population of interest for the variable $Y$ has a size of $N$. Select a sample size of $(2m)^3$ from the population. Divide them randomly into $2m$ RSS, each with $(2m)^2$. Apply the RSS technique to each set to produce $2m$ RSS each with size $2m$. To draw a DEMRSS of size $2m$, apply the EMRSS technique to $2m$ sets once a cycle is completed. To get a sample with the size $n = 2rm$, repeat the method $r$ times. The following two layouts clearly explain the newly proposed selection procedure.

**Layout for even $m$:** To illuminate DEMRSS with even size, for simplicity, we consider $m = 2$, $(2m)^3 = 64$, $(2m)^2 = 16$, $r = 2$ and $n = 2rm = 8$. The sampling layout is described in the following steps.

**Step 1:** *Create sets. For example, the set 1 is obtained as follows:*

$$Set1 = \begin{bmatrix} Y^{(1)}_{1(1:4)1} & Y^{(1)}_{1(2:4)1} & Y^{(1)}_{1(3:4)1} & Y^{(1)}_{1(4:4)1} \\ Y^{(1)}_{2(1:4)1} & Y^{(1)}_{2(2:4)1} & Y^{(1)}_{2(3:4)1} & Y^{(1)}_{2(4:4)1} \\ Y^{(1)}_{3(1:4)1} & Y^{(1)}_{3(2:4)1} & Y^{(1)}_{3(3:4)1} & Y^{(1)}_{3(4:4)1} \\ Y^{(1)}_{4(1:4)1} & Y^{(1)}_{4(2:4)1} & Y^{(1)}_{4(3:4)1} & Y^{(1)}_{4(4:4)1} \end{bmatrix}$$

Repeat it in the other three sets as well.

**Step 2:**

$$\begin{bmatrix} Y^{(1)(1:4)}_{1(1:4)1} & Y^{(1)(2:4)}_{2(2:4)1} & Y^{(1)(3:4)}_{3(3:4)1} & Y^{(1)(4:4)}_{4(4:4)1} & from\ Set\ 1 \\ Y^{(2)(1:4)}_{1(1:4)1} & Y^{(2)(2:4)}_{2(2:4)1} & Y^{(2)(3:4)}_{3(3:4)1} & Y^{(2)(4:4)}_{4(4:4)1} & from\ Set\ 2 \\ Y^{(3)(1:4)}_{1(1:4)1} & Y^{(3)(2:4)}_{2(2:4)1} & Y^{(3)(3:4)}_{3(3:4)1} & Y^{(3)(4:4)}_{4(4:4)1} & from\ Set\ 3 \\ Y^{(4)(1:4)}_{1(1:4)1} & Y^{(4)(2:4)}_{2(2:4)1} & Y^{(4)(3:4)}_{3(3:4)1} & Y^{(4)(4:4)}_{4(4:4)1} & from\ Set\ 4 \end{bmatrix}$$

So, from the first cycle, we get $Y^{(1)(1:4)}_{1(1:4)1}$, $Y^{(2)(4:4)}_{4(4:4)1}$, $Y^{(3)(2:4)}_{2(2:4)1}$ and $Y^{(4)(3:4)}_{3(3:4)1}$. Along the same lines, from the second cycle, we will select $Y^{(1)(1:4)}_{1(1:4)2}$, $Y^{(2)(4:4)}_{4(4:4)2}$, $Y^{(3)(2:4)}_{2(2:4)2}$ and $Y^{(4)(3:4)}_{3(3:4)2}$. From step 2 of the cycle, the underlined units are chosen by using the DEMRSS procedure.

**Layout for odd $m$:** To illuminate DEMRSS with odd size, for simplicity, we consider $m = 3$, $(2m)^3 = 216$, $(2m)^2 = 36$, $r = 2$ and $n = 2rm = 12$. The sampling layout is as under

*Step 1*: *Create sets. For example, the set 1 is obtained as follows*:

$$Set\ 1 = \begin{bmatrix} Y_{1(1:6)1}^{(1)} & Y_{1(2:6)1}^{(1)} & Y_{1(3:6)1}^{(1)} & Y_{1(4:6)1}^{(1)} & Y_{1(5:6)1}^{(1)} & Y_{1(6:6)1}^{(1)} \\ Y_{2(1:6)1}^{(1)} & Y_{2(2:6)1}^{(1)} & Y_{2(3:6)1}^{(1)} & Y_{2(4:6)1}^{(1)} & Y_{2(5:6)1}^{(1)} & Y_{2(6:6)1}^{(1)} \\ Y_{3(1:6)1}^{(1)} & Y_{3(2:6)1}^{(1)} & Y_{3(3:6)1}^{(1)} & Y_{3(4:6)1}^{(1)} & Y_{3(5:6)1}^{(1)} & Y_{3(6:6)1}^{(1)} \\ Y_{4(1:6)1}^{(1)} & Y_{4(2:6)1}^{(1)} & Y_{4(3:6)1}^{(1)} & Y_{4(4:6)1}^{(1)} & Y_{4(5:6)1}^{(1)} & Y_{4(6:6)1}^{(1)} \\ Y_{5(1:6)1}^{(1)} & Y_{5(2:6)1}^{(1)} & Y_{5(3:6)1}^{(1)} & Y_{5(4:6)1}^{(1)} & Y_{5(5:6)1}^{(1)} & Y_{5(6:6)1}^{(1)} \\ Y_{6(1:6)1}^{(1)} & Y_{6(2:6)1}^{(1)} & Y_{6(3:6)1}^{(1)} & Y_{6(4:6)1}^{(1)} & Y_{6(5:6)1}^{(1)} & Y_{6(6:6)1}^{(1)} \end{bmatrix}$$

Repeat it in other five sets as well.

*Step 2*:

$$\begin{bmatrix} Y_{1(1:6)1}^{(1)(1:6)} & Y_{2(2:6)1}^{(1)(2:6)} & Y_{3(3:6)1}^{(1)(3:6)} & Y_{4(4:6)1}^{(1)(4:6)} & Y_{5(5:6)1}^{(1)(5:6)} & Y_{6(6:6)1}^{(1)(6:6)} & from\ Set\ 1 \\ Y_{1(1:6)1}^{(2)(1:6)} & Y_{2(2:6)1}^{(2)(2:6)} & Y_{3(3:6)1}^{(250)} & Y_{4(4:6)1}^{(2)} & Y_{5(5:6)1}^{(2)(5:6)} & Y_{6(6:6)1}^{(2)(6:6)} & from\ Set\ 2 \\ Y_{1(1:6)1}^{(3)(1:6)} & Y_{2(2:6)1}^{(3)(2:6)} & Y_{3(3:6)1}^{(3)(3:6)} & Y_{4(4:6)1}^{(3)(4:6)} & Y_{5(5:6)1}^{(3)(5:6)} & Y_{6(6:6)1}^{(3)(6:6)} & from\ Set\ 3 \\ Y_{1(1:6)1}^{(4)(1:6)} & Y_{2(2:6)1}^{(4)(2:6)} & Y_{3(3:6)1}^{(4)(3:6)} & Y_{4(4:6)1}^{(4)} & Y_{5(5:6)1}^{(4)(5:6)} & Y_{6(6:6)1}^{(4)(6:6)} & from\ Set\ 4 \\ Y_{1(1:6)1}^{(5)(1:6)} & Y_{2(2:6)1}^{(5)(2:6)} & Y_{3(3:6)1}^{(5)(3:6)} & Y_{4(4:6)1}^{(5)(4:6)} & Y_{5(5:6)1}^{(5)(5:6)} & Y_{6(6:6)1}^{(5)(6:6)} & from\ Set\ 5 \\ Y_{1(1:6)1}^{(6)(1:6)} & Y_{2(2:6)1}^{(6)(2:6)} & Y_{3(3:6)1}^{(6)(3:6)} & Y_{4(4:6)1}^{(6)(4:6)} & Y_{5(5:6)1}^{(6)(5:6)} & Y_{6(6:6)1}^{(6)(6:6)} & from\ Set\ 6 \end{bmatrix}$$

So, from the first cycle, we got $Y_{1(1:6)1}^{(1)(1:6)}$, $Y_{6(6:6)1}^{(2)(6:6)}$, $Y_{3(3:6)1}^{(3)(3:6)}$, $Y_{4(4:6)1}^{(4)(4:6)}$, $Y_{3(3:6)1}^{(5)(3:6)}$ and $Y_{4(4:6)1}^{(6)(4:6)}$. On the same lines, from the second cycle, we will select $Y_{1(1:6)2}^{(1)(1:6)}$, $Y_{6(6:6)2}^{(2)(6:6)}$, $Y_{3(3:6)2}^{(3)(3:6)}$, $Y_{4(4:6)2}^{(4)(4:6)}$, $Y_{3(3:6)2}^{(5)(3:6)}$ and $Y_{4(4:6)2}^{(6)(4:6)} Y_{4(4:6)2}^{(6)(4:6)}$. From step 2 of the cycle, the underlined units are chosen by using the DEMRSS procedure.

The mean estimators under DEMRSS for even (*E*) and odd (*O*) sample sizes are given as follows:

$$\widehat{\mu}_{y,DEMRSS}^{(E)} = \frac{1}{2rm} \sum_{j=1}^{r} \left[ \begin{array}{c} \left( \sum_{i=1}^{\frac{m}{2}} Y_{1(1)j}^{i(1)} + \sum_{i=\frac{m}{2}+1}^{m} Y_{2m(2m)j}^{i(2m)} \right) + \\ \left( \sum_{i=m+1}^{\frac{3m}{2}} Y_{m(m)j}^{i(m)} + \sum_{i=\frac{3m}{2}+1}^{2m} Y_{(m+1)(m+1)j}^{i(m+1)} \right) \end{array} \right], \tag{10}$$

$$\widehat{\mu}_{y,DEMRSS}^{(O)} = \frac{1}{2rm} \sum_{j=1}^{r} \left[ \begin{array}{c} \left( \sum_{i=1}^{\frac{(m-1)}{2}} Y_{1(1)j}^{i(1)} + \sum_{i=\left\{\frac{(m-1)}{2}\right\}+1}^{m-1} Y_{2m(2m)j}^{i(2m)} + Y_{m(m)j}^{m(m)} \right) \\ + \left( Y_{(m+1)(m+1)j}^{(m+1)(m+1)} + \sum_{i=m+2}^{\frac{(3m+1)}{2}} Y_{m(m)j}^{i(m)} + \sum_{i=\frac{3(m+1)}{2}}^{2m} Y_{(m+1)(m+1)j}^{i(m+1)} \right) \end{array} \right]. \tag{11}$$

Both the estimators $\widehat{\mu}_{y,DEMRSS}^{(E)}$ and $\widehat{\mu}_{y,DEMRSS}^{(O)}$ are unbiased if the underlying distribution is

symmetric about mean see Ahmed and Shabbir [9] and the variances are respectively.

$$var\left(\widehat{\mu}_{y,DEMRSS}^{(E)}\right) = \left[\frac{\sigma^2}{2rm} - \frac{1}{4rm^2}\left\{\frac{m}{2}\left(\varphi_{1(1)}^{(1)^2} + \varphi_{2m(2m)}^{(2m)^2}\right) + \frac{m}{2}\left(\varphi_{m(m)}^{(m)^2} + \varphi_{(m+1)(m+1)}^{(m+1)^2}\right)\right\}\right]. \quad (12)$$

$$var\left(\widehat{\mu}_{y,DEMRSS}^{(O)}\right) = \left[\frac{\sigma^2}{2rm} - \frac{1}{4rm^2}\left\{\frac{m-1}{2}\left(\varphi_{1(1)}^{(1)^2} + \varphi_{2m(2m)}^{(2m)^2}\right) + \frac{m+1}{2}\left(\varphi_{m(m)}^{(m)^2} + \varphi_{(m+1)(m+1)}^{(m+1)^2}\right)\right\}\right], \quad (13)$$

where, $\varphi_{i(i)}^{(i)^2} = \left(\mu_{y(i)} - \mu_{(y)}\right)^2$ and $\mu_{y(i)}$ denotes population mean of $i^{\text{th}}$ order statistic for ($i = 1, 2, 3, \ldots, 2m$).

## 3. Design of simulation study

For the efficiency comparison of the sample mean utilizing SRS, RSS, DRSS, EMRSS, and DEMRSS strategies, a Monte Carlo simulation is carried out with the help of R-Studio. Further, the percent relative efficiency (PRE) is considered as a performance criterion.

### 3.1. Algorithm for the simulation study

The stepwise procedure adopted is given below:

1. A hypothetical population of size $N = 1000$ for concomitant variable $X$ is generated using the following distributions:

   Symmetric distributions i) Uniform (0, 1) ii) Normal (5, 1)
   Asymmetric distributions i) Gamma (4, 3) ii) Weibull(1.5, 5)

2. Using the regression model $Y = X+e$, where '$e$' is the error term such that $e \sim N(0,1)$, the study variable $Y$ is computed.

3. The number of iterations is considered equal to 100,000.

4. The performance of estimators (1), (3), (6), (7), (10) and (11) has been calculated using a set size of $m$ (3 to 10) and a number of cycles $r = 5$.

5. For comparison purposes, both perfect (ranking with respect to study variable $Y$) and imperfect (ranking with respect to concomitant variable X) rankings have been used.

6. Using the following equations, estimators' percent relative efficiencies (PREs) have been calculated for symmetric and asymmetric distributions.

   For unbiased estimators : $PRE = \dfrac{var\left(\widehat{\mu}_{y,SRS}\right)}{var\left(\widehat{\mu}_{y,k}\right)} \times 100,$ (14)

   For biased estimators : $PRE = \dfrac{var\left(\widehat{\mu}_{y,SRS}\right)}{MSE\left(\widehat{\mu}_{y,k}\right)} \times 100,$ (15)

where $var\left(\widehat{\mu}_{y,SRS}\right) = \frac{\sigma^2}{n}$, $MSE\left(\widehat{\mu}_{y,k}\right)$ is the mean square error of estimators and '$k$' denotes RSS, DRSS, EMRSS and DEMRSS.

## 4. Results and discussion

Fig 1 shows the simulation outcomes for mean estimation under perfect ranking. Results indicate that as sample size increases, PREs for all estimators rise. The results also showed that the PREs of mean estimators based on DEMRSS are higher for all distributions and sample sizes than those of its rival estimators. Under the Weibull distribution, where its maximum PRE is 488.98, DEMRSS yields more efficient results. The greatest PRE for DEMRSS with a uniform distribution is 268.69. In contrast, the highest PREs of DEMRSS for Normal and Gamma distributions are 404.26 and 222.53, respectively.

Results for mean estimation under imperfect ranking are shown in Fig 2. Results show that in imperfect ranking, PREs of all estimators rise with increasing sample size. For imperfect ranking, the PREs of DEMRSS are also higher than those of its rival estimators for all distributions and sample sizes, as we have seen in the case of perfect ranking. With a maximum PRE of 244.16 and a uniform distribution, DEMRSS yields more efficient outcomes. Maximum PRE for DEMRSS under Weibull distribution is 223.36. In comparison, the highest PREs of DEMRSS for Normal and Gamma distributions are 208.14 and 158.88, respectively.

Results demonstrate that in both symmetric and asymmetric distributions, PREs of all estimators drop under imperfect ranking compared to perfect ranking. Maximum PREs of DEMRSS for perfect and imperfect ranking in Uniform distribution are 268.69 and 244.16, respectively, whereas maximum PREs of DEMRSS for perfect and imperfect ranking in Normal distribution are 404.26 and 208.14, respectively. Furthermore, the highest PREs of DEMRSS for perfect and imperfect ranking in the Gamma distribution are 222.53 and 158.88,

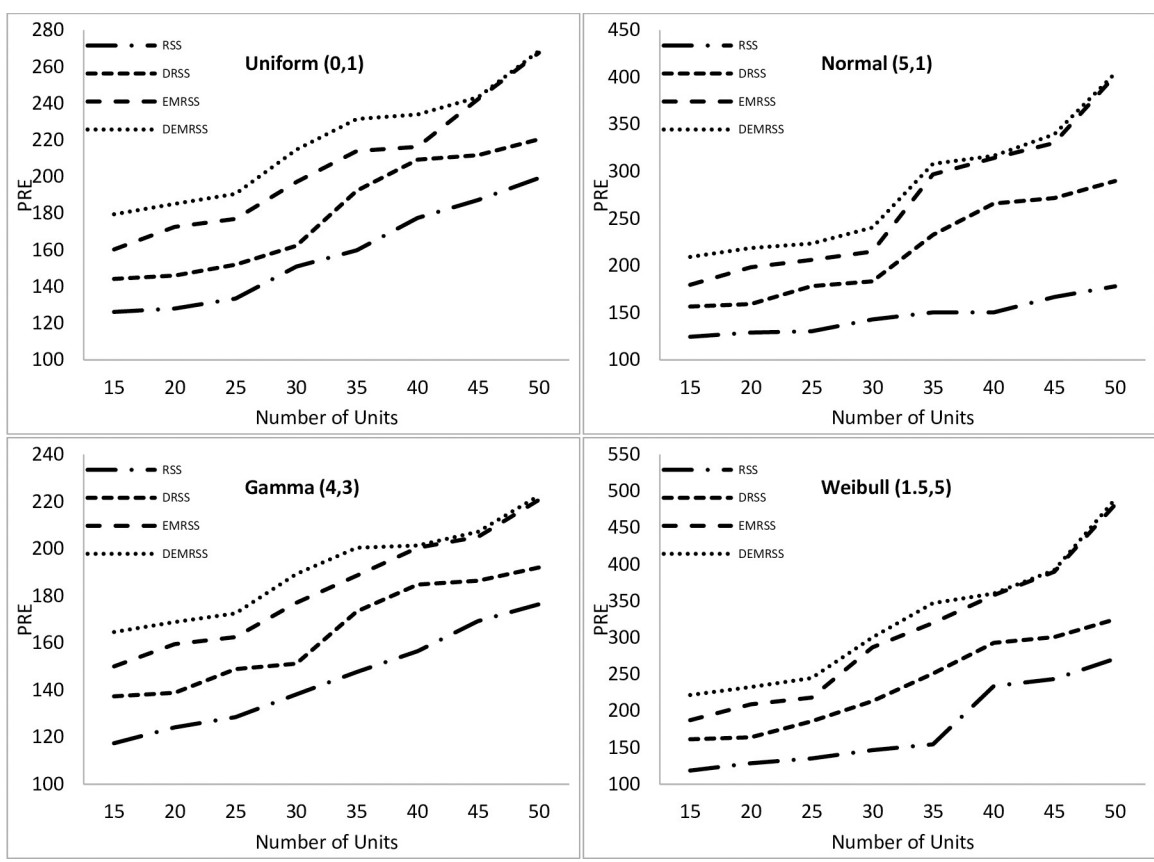

**Fig 1. PREs of mean estimators with respect to SRS under perfect ranking.**

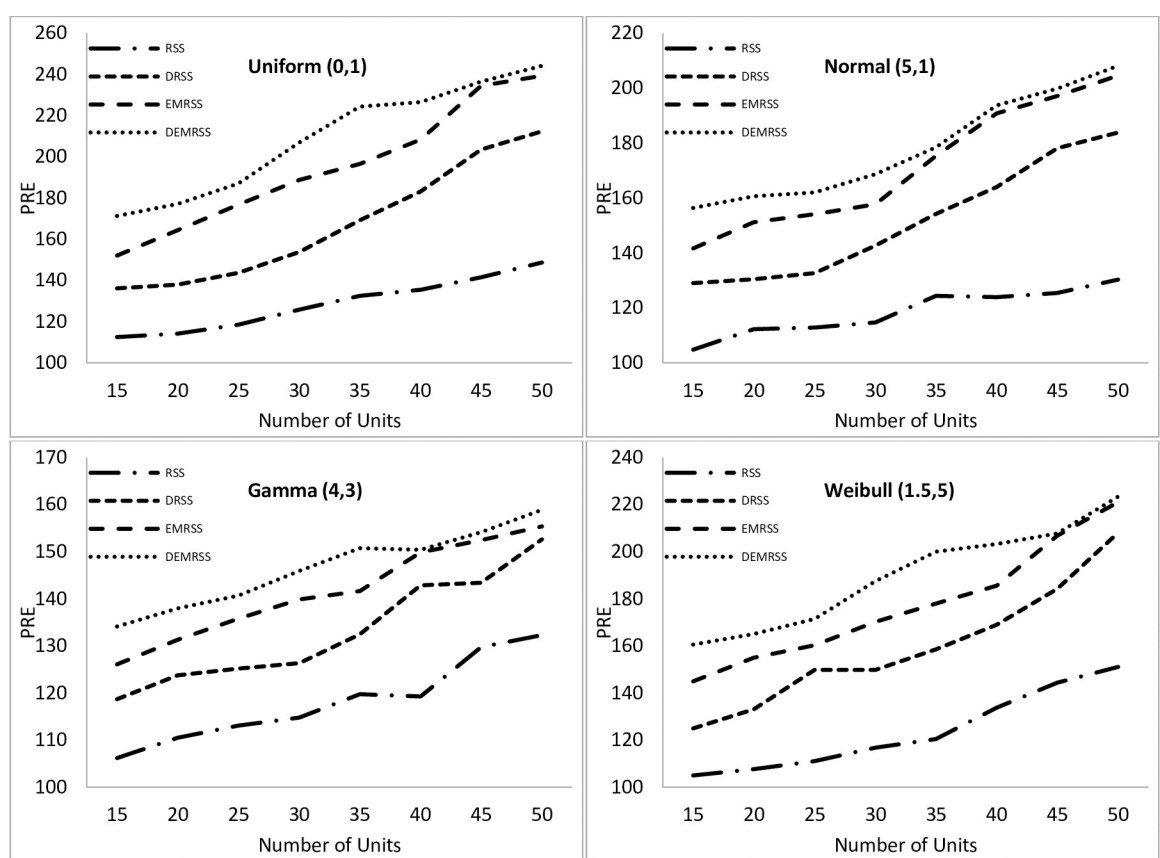

**Fig 2. PREs of mean estimators with respect to SRS under imperfect ranking.**

respectively, whereas the maximum PREs of DEMRSS for perfect and imperfect ranking in the Weibull distribution are 488.98 and 223.36, respectively.

However, for perfect ranking, the percent relative efficiency under the Weibull distribution is clearly on the upper side. For imperfect ranking, PREs of DEMRSS under the Weibull distribution are marginally higher than the Normal distribution.

## 4.1. Real-life examples

A real-world data set provided by MICS (2018–19) is used to examine the percent relative efficiencies (PREs) of proposed and existing sampling schemes. Let $Y$ be the weight (Kg), and $X$ be the height (cm) of the individuals in the population of children. Consider a random sample of $n = 30$ with $m = 10$ and $r = 3$ to be selected to estimate the mean weight of the population. The following parameters are computed based on these two variables. Using the Easy Fit Version software, it is found that both study variable $Y$ and auxiliary variable X follow the Weibull distribution. The values of the shape, scale, and location parameters were computed for the study variable (weight) and are 3.145, 12.049, and 0.000, respectively. The auxiliary variable (height) also follows the Weibull distribution, and calculated parametric values are 6.806, 88.322, and 0.000, respectively.

$$N = 39118, \quad \mu_x = 94.6221, \quad \mu_y = 12.1883,$$
$$C_x = 1.0678, \quad C_y = 0.8853, \quad C_{xy} = 0.7016,$$
$$\rho_{xy} = 0.7421,$$

**Table 1. PREs of mean estimators in real-life data with respect to SRS under perfect ranking.**

| R | m | N | RSS | DRSS | EMRSS | DEMRSS |
|---|---|---|---|---|---|---|
| 1 | 3 | 3 | 101.49 | 103.93 | 104.52 | **105.77** |
| | 4 | 4 | 104.55 | 107.89 | 108.92 | **111.90** |
| | 5 | 5 | 103.18 | 105.91 | 109.45 | **111.60** |
| | 6 | 6 | 101.09 | 108.21 | 110.24 | **112.97** |
| | 7 | 7 | 104.67 | 111.48 | 115.19 | **119.55** |
| | 8 | 8 | 104.89 | 112.55 | 116.87 | **121.20** |
| | 9 | 9 | 105.26 | 114.23 | 119.30 | **121.34** |
| | 10 | 10 | 104.12 | 111.91 | 118.95 | **122.90** |
| 2 | 3 | 6 | 100.46 | 103.12 | 105.04 | **106.44** |
| | 4 | 8 | 102.16 | 105.22 | 108.60 | **110.44** |
| | 5 | 10 | 101.28 | 105.84 | 110.64 | **113.09** |
| | 6 | 12 | 101.22 | 108.23 | 111.62 | **113.09** |
| | 7 | 14 | 103.76 | 111.42 | 115.61 | **120.58** |
| | 8 | 16 | 105.56 | 110.91 | 117.55 | **122.52** |
| | 9 | 18 | 104.42 | 116.47 | 120.37 | **122.74** |
| | 10 | 20 | 103.10 | 115.80 | 122.21 | **126.99** |
| 3 | 3 | 9 | 100.40 | 101.54 | 102.75 | **102.86** |
| | 4 | 12 | 101.53 | 102.84 | 105.03 | **105.36** |
| | 5 | 15 | 100.67 | 101.91 | 103.92 | **106.12** |
| | 6 | 18 | 101.03 | 106.89 | 109.69 | **110.89** |
| | 7 | 21 | 103.15 | 109.47 | 112.85 | **116.82** |
| | 8 | 24 | 102.14 | 106.30 | 114.31 | **117.65** |
| | 9 | 27 | 103.66 | 113.35 | 116.41 | **118.25** |
| | 10 | 30 | 102.55 | 112.73 | 116.00 | **119.54** |
| 4 | 3 | 12 | 100.40 | 102.67 | 104.30 | **105.49** |
| | 4 | 16 | 101.85 | 104.45 | 107.31 | **108.84** |
| | 5 | 20 | 101.09 | 104.96 | 108.97 | **111.00** |
| | 6 | 24 | 101.03 | 106.89 | 109.69 | **110.89** |
| | 7 | 28 | 103.15 | 110.75 | 112.85 | **116.82** |
| | 8 | 32 | 104.62 | 110.39 | 116.05 | **118.23** |
| | 9 | 36 | 103.66 | 114.92 | 117.07 | **118.25** |
| | 10 | 40 | 102.55 | 114.28 | 120.71 | **121.34** |

The findings in Table 1 reveal the higher PREs values for the DERMSS strategy under the perfect ranking, which is evidence that the mean estimator under DEMRSS is more accurate than all other estimators that were considered for this investigation. The best performance has been seen at $n = 20$ with $m = 10$ and $r = 2$, i.e., 126.99.

Table 2 illustrates the percentage relative efficiency of the mean estimator for both the suggested and current sampling methods when imperfect ranking is employed. Height is a concomitant variable and has been utilized for the ranking process. Results demonstrate that the mean estimate of the DEMRSS outperforms all other estimators in an imperfect ranking. The best performance has been seen at $n = 32$ with $m = 8$ and $r = 4$, *i.e.*, 126.99. The results also showed that in the imperfect ranking situation compared to perfect ranking, there is a downward tendency in the overall relative efficiency of all estimators due to errors of ranking.

**Table 2. PREs of mean estimators in real-life data with respect to SRS under imperfect ranking.**

| r | m | N | RSS | DRSS | EMRSS | DEMRSS |
|---|---|---|---|---|---|---|
| **1** | 3 | 3 | 100.35 | 101.33 | 101.99 | **102.39** |
| | 4 | 4 | 100.50 | 101.28 | 101.84 | **102.38** |
| | 5 | 5 | 100.15 | 101.87 | 102.97 | **103.99** |
| | 6 | 6 | 100.40 | 102.81 | 103.50 | **105.02** |
| | 7 | 7 | 101.96 | 104.19 | 104.94 | **105.93** |
| | 8 | 8 | 101.43 | 104.59 | 106.17 | **107.77** |
| | 9 | 9 | 101.47 | 102.17 | 103.31 | **104.57** |
| | 10 | 10 | 101.55 | 102.91 | 104.59 | **106.49** |
| **2** | 3 | 6 | 100.59 | 102.62 | 103.51 | **104.17** |
| | 4 | 8 | 100.74 | 102.57 | 103.35 | **104.17** |
| | 5 | 10 | 100.39 | 103.21 | 104.59 | **105.94** |
| | 6 | 12 | 100.65 | 104.24 | 105.18 | **107.09** |
| | 7 | 14 | 102.29 | 105.74 | 106.76 | **108.09** |
| | 8 | 16 | 101.74 | 106.21 | 108.13 | **110.16** |
| | 9 | 18 | 101.79 | 103.65 | 105.08 | **106.72** |
| | 10 | 20 | 101.88 | 104.46 | 106.51 | **108.86** |
| **3** | 3 | 9 | 100.76 | 102.62 | 103.55 | **104.69** |
| | 4 | 12 | 100.89 | 102.58 | 103.42 | **104.69** |
| | 5 | 15 | 100.60 | 103.11 | 104.43 | **106.16** |
| | 6 | 18 | 100.82 | 103.95 | 104.91 | **107.10** |
| | 7 | 21 | 102.16 | 105.16 | 106.19 | **107.92** |
| | 8 | 24 | 101.71 | 105.52 | 107.28 | **109.58** |
| | 9 | 27 | 101.75 | 103.45 | 104.82 | **106.80** |
| | 10 | 30 | 101.82 | 104.11 | 105.96 | **108.53** |
| **4** | 3 | 12 | 102.08 | 104.13 | 106.45 | **107.95** |
| | 4 | 16 | 102.19 | 104.10 | 106.34 | **107.96** |
| | 5 | 20 | 101.97 | 104.54 | 107.22 | **109.23** |
| | 6 | 24 | 102.16 | 105.24 | 107.64 | **110.04** |
| | 7 | 28 | 103.25 | 106.23 | 108.72 | **110.74** |
| | 8 | 32 | 102.90 | 106.53 | 109.66 | **112.16** |
| | 9 | 36 | 102.96 | 104.89 | 107.68 | **109.92** |
| | 10 | 40 | 103.03 | 105.43 | 108.64 | **111.37** |

## 5. Conclusion and future directions

In this study, a method to estimate population mean using ranked set sampling has been developed. It is effective and simple to use. DRSS and EMRSS are combined in the suggested technique. The comparison of the percent relative efficiency of the DEMRSS with the SRS, RSS, DRSS, and EMRSS has been made using extensive Monte Carlo simulation, taking into account the Uniform, Normal, Gamma, and Weibull distributions. Based on the results of a Monte Carlo simulation, it can be concluded that for all distributions and sample sizes taken into consideration, DEMRSS provides a more efficient mean estimator than SRS, RSS, DRSS, and EMRSS. The best performance is observed under the Weibull distribution, which has a perfect ranking for all sample sizes. Furthermore, a real-world data set provided by MICS (2018–19) has been utilized to show how much more effective DEMRSS is compared to its rival techniques. The practical results have been observed to align closely with the outcomes produced by the Monte Carlo simulation mentioned earlier. The proposed ranked set

sampling technique proves useful in research fields where ranking all observations is laborious, but identifying the median and extreme values is straightforward. Future studies should suggest some generalized ratio, product, regression, and exponential estimators using DEMRSS for estimating population parameters in varied environments. This proposal will also provide a cost-effective solution for designing the control charts like Shewhart, EWMA, and CUSUM to monitor location parameter.

## Supporting information

**S1 Appendix.**
(DOCX)

## Acknowledgments

Thank you to the anonymous reviewers and associate editors for their insightful comments and input, which significantly enhanced the quality of the article.

## Author Contributions

**Conceptualization:** Muhammad Zubair, Asad Ali.

**Data curation:** Fathia Moh. Al Samman.

**Formal analysis:** Asad Ali, Mohammed M. A. Almazah.

**Funding acquisition:** Afrah Al-Bossly.

**Investigation:** Mohammed M. A. Almazah, Kanwal Iqbal.

**Methodology:** Muhammad Zubair, Fathia Moh. Al Samman.

**Project administration:** Kanwal Iqbal.

**Resources:** Afrah Al-Bossly.

**Software:** Asad Ali.

**Supervision:** Seyab Yasin.

**Visualization:** Afrah Al-Bossly.

**Writing – original draft:** Asad Ali.

**Writing – review & editing:** Asad Ali, Kanwal Iqbal.

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
