## [Decision Letter · Decision Letter 0]

1 Apr 2024

PONE-D-24-07535Double Extreme-Cum-Median Ranked Set SamplingPLOS ONE

Dear Dr. Iqbal,

Thank you for submitting your manuscript to PLOS ONE. After careful consideration, we feel that it has merit but does not fully meet PLOS ONE’s publication criteria as it currently stands. Therefore, we invite you to submit a revised version of the manuscript that addresses the points raised during the review process.

We look forward to receiving your revised manuscript.

Kind regards,

Yu Zhou

Academic Editor

PLOS ONE

Journal Requirements:

"Their authors extend Their appreciation to the Deanship of Scientific Research at King Khalid University for funding this work through Large Research Groups Project under grant number (RGP.2/41/45) and this study is supported via funding from Prince Sattam bin Abdulaziz University project number (PSAU/2023/R/1444)."

"Their authors extend Their appreciation to the Deanship of Scientific Research at King Khalid University for funding this work through Large Research Groups Project under grant number (RGP.2/41/45) and this study is supported via funding from Prince Sattam bin Abdulaziz University project number (PSAU/2023/R/1444)."

"Their authors extend Their appreciation to the Deanship of Scientific Research at King Khalid University for funding this work through Large Research Groups Project under grant number (RGP.2/41/45) and this study is supported via funding from Prince Sattam bin Abdulaziz University project number (PSAU/2023/R/1444)."

6. n this instance it seems there may be acceptable restrictions in place that prevent the public sharing of your minimal data. However, in line with our goal of ensuring long-term data availability to all interested researchers, PLOS’ Data Policy states that authors cannot be the sole named individuals responsible for ensuring data access (http://journals.plos.org/plosone/s/data-availability#loc-acceptable-data-sharing-methods).

7. PLOS requires an ORCID iD for the corresponding author in Editorial Manager on papers submitted after December 6th, 2016. Please ensure that you have an ORCID iD and that it is validated in Editorial Manager. To do this, go to ‘Update my Information’ (in the upper left-hand corner of the main menu), and click on the Fetch/Validate link next to the ORCID field. This will take you to the ORCID site and allow you to create a new iD or authenticate a pre-existing iD in Editorial Manager. Please see the following video for instructions on linking an ORCID iD to your Editorial Manager account: https://www.youtube.com/watch?v=_xcclfuvtxQ

Reviewers' comments:

Reviewer's Responses to Questions

**Comments to the Author**

1. Is the manuscript technically sound, and do the data support the conclusions?

Reviewer #1: Yes

Reviewer #2: Yes

2. Has the statistical analysis been performed appropriately and rigorously? 

Reviewer #1: Yes

Reviewer #2: Yes

3. Have the authors made all data underlying the findings in their manuscript fully available?

Reviewer #1: Yes

Reviewer #2: Yes

4. Is the manuscript presented in an intelligible fashion and written in standard English?

Reviewer #1: No

Reviewer #2: Yes

5. Review Comments to the Author

Reviewer #1: "Comments to the Authors"

“Double Extreme-Cum-Median Ranked Set Sampling”

Ref. No: PONE-D-24-07535

This study proposes the Double Extreme-cum-Median Ranked Set (DEMRSS) sampling to provide mean estimation when the study variable has heterogeneity and outliers (extreme values). A simulation study is designed to discuss the performance of the proposed estimator and also for the comparative study among existing ones. Later, a real-data set application is used to highlight the implementation of the stated proposal. The article is well written, but the outcomes of this research are significant and beneficial for practitioners to use in the future. I would be happy to reconsider my decision only after the authors consider the following comments/suggestions and revise the manuscript accordingly.

1) The given literature review is not adequate. Therefore, it is suggested that more studies be added. Specifically, integrating ranked set samplings with SPC techniques is a hot topic. Consequently, I may suggest adding a specific paragraph on the importance or usage of ranked set samplings in control chart studies.

2) The authors provided the references 6 to 25 without giving details of them. Please write the significance of each citation in at least one line.

3) The citation style needs to be improved. For example, wherever a line starts with a reference, then reference can be made in the form of Author [number]; a similar form may also follow when a citation is at the end of the line but with the word “by”. For example, ranked set sampling is proposed by Zubair et al. [1].

4) Sections are not adequately divided. I may suggest making sections as (1) Introduction; (2) Mathematical Framework (including ranked set sampling (RSS), Double RSS, Extreme-cum-Median RSS, Proposed sampling Scheme); (3) Design of Simulation Study (including Design parameters for simulations, algorithm of the simulations, Performance Measures); (4) Results and Discussion; (5) Real-life examples and (6) Conclusion and Future Directions.

5) Provide the algorithm of the simulation study.

6) Provide a graphical display of how RSS, DRSS, EMRSS, and DEMRSS sampling schemes differ from each other.

7) It is suggested that the manuscript be read and the grammatical errors resolved. Moreover, use the consistent terms. For example, “Weibull distribution” or “Weibull Distribution”

Reviewer #2: 1- Abstract needs a revision, please don’t give abbreviations in abstract and avoid to write in brackets.

2- Please don’t cite so many references together without discussing them as you have given after equation (2).

3- Give a separate table of list of abbreviations and notations. This can be placed in appendix.

4- Literature needs to be arranged in the different sections. According to the heading. Please consider the following papers and cite them in the different sections accordingly.

Noor-ul-Amin, M., Arif, F. & Hanif, M. Modified Extreme Ranked Sets Sampling with Auxiliary Variable. Proc. Natl. Acad. Sci., India, Sect. A Phys. Sci. 91, 537–542 (2021). https://doi.org/10.1007/s40010-020-00698-6

Noor-ul-Amin, M., Tayyab, M. Influence on Estimation of Mean by Using Quartile Paired Double Ranked Set Sampling. Proc. Natl. Acad. Sci., India, Sect. A Phys. Sci. 91, 117–121 (2021). https://doi.org/10.1007/s40010-020-00680-2

Tayyab, M., Noor-ul-Amin, M. & Hanif, M. Quartile Pair Ranked Set Sampling: Development and Estimation. Proc. Natl. Acad. Sci., India, Sect. A Phys. Sci. 91, 111–116 (2021). https://doi.org/10.1007/s40010-019-00651-2

6. PLOS authors have the option to publish the peer review history of their article (what does this mean?). If published, this will include your full peer review and any attached files.

Reviewer #1: No

Reviewer #2: No

---

## [Author Response · Author response to Decision Letter 0]

24 Apr 2024

Replies to Referee’s Comments

Double Extreme-Cum-Median Ranked Set Sampling

We are thankful to the Editor and the anonymous Reviewers for the useful comments and the opportunity to improve our paper. We have revised the article by addressing all the suggested points. Here are the point-by-point responses to all the comments. 

Reviewer #1: 

1) The given literature review is not adequate. Therefore, it is suggested that more studies be added. Specifically, integrating ranked set samplings with SPC techniques is a hot topic. Consequently, I may suggest adding a specific paragraph on the importance or usage of ranked set samplings in control chart studies.

Reply: Thanks for your suggestion. The importance or usage of ranked set samplings in control chart studies is added. 

2) The authors provided the references 6 to 25 without giving details of them. Please write the significance of each citation in at least one line.

Reply: Thanks for your suggestion. The details of references are added. 

3) The citation style needs to be improved. For example, wherever a line starts with a reference, then reference can be made in the form of Author [number]; a similar form may also follow when a citation is at the end of the line but with the word “by”. For example, ranked set sampling is proposed by Zubair et al. [1].

Reply: Thanks for your suggestion. The citation style is improved. 

4) Sections are not adequately divided. I may suggest making sections as (1) Introduction; (2) Mathematical Framework (including ranked set sampling (RSS), Double RSS, Extreme-cum-Median RSS, Proposed sampling Scheme); (3) Design of Simulation Study (including Design parameters for simulations, algorithm of the simulations, Performance Measures); (4) Results and Discussion; (5) Real-life examples and (6) Conclusion and Future Directions.

Reply: Thanks for your suggestion. Sections are properly divided. 

5) Provide the algorithm of the simulation study.

Reply: Thanks for your suggestion. Simulation Procedure is provided in detail. 

6) Provide a graphical display of how RSS, DRSS, EMRSS, and DEMRSS sampling schemes differ from each other.

Reply: Thanks for your suggestion. Provided in Fig 1 & 2.

7) It is suggested that the manuscript be read and the grammatical errors resolved. Moreover, use the consistent terms. For example, “Weibull distribution” or “Weibull Distribution”

Reply: Thank you for the suggestion. We have carefully checked the language and typo errors in the article.

Reviewer #2: 

1- Abstract needs a revision, please don’t give abbreviations in abstract and avoid to write in brackets.

Reply: Thanks for your suggestions. We have revised the abstract carefully. 

2- Please don’t cite so many references together without discussing them as you have given after equation (2).

Reply: Thanks for your suggestions. The said changes are made. 

3- Give a separate table of list of abbreviations and notations. This can be placed in appendix.

Reply: Thanks for your suggestions. Abbreviations and notations are already explained in the manuscript.

4- Literature needs to be arranged in the different sections. According to the heading. Please consider the following papers and cite them in the different sections accordingly.

Noor-ul-Amin, M., Arif, F. & Hanif, M. Modified Extreme Ranked Sets Sampling with Auxiliary Variable. Proc. Natl. Acad. Sci., India, Sect. A Phys. Sci. 91, 537–542 (2021). https://doi.org/10.1007/s40010-020-00698-6

Noor-ul-Amin, M., Tayyab, M. Influence on Estimation of Mean by Using Quartile Paired Double Ranked Set Sampling. Proc. Natl. Acad. Sci., India, Sect. A Phys. Sci. 91, 117–121 (2021). https://doi.org/10.1007/s40010-020-00680-2

Tayyab, M., Noor-ul-Amin, M. & Hanif, M. Quartile Pair Ranked Set Sampling: Development and Estimation. Proc. Natl. Acad. Sci., India, Sect. A Phys. Sci. 91, 111–116 (2021). https://doi.org/10.1007/s40010-019-00651-2

Reply: Thanks for your suggestion. We have cited the said references in the revised manuscript.

---

## [Decision Letter · Decision Letter 1]

16 Aug 2024

PONE-D-24-07535R1Double Extreme-Cum-Median Ranked Set SamplingPLOS ONE

Dear Dr. Iqbal,

Thank you for submitting your manuscript to PLOS ONE. After careful consideration, we feel that it has merit but does not fully meet PLOS ONE’s publication criteria as it currently stands. Therefore, we invite you to submit a revised version of the manuscript that addresses the points raised during the review process.

We look forward to receiving your revised manuscript.

Kind regards,

Yu Zhou

Academic Editor

PLOS ONE

Reviewers' comments:

Reviewer's Responses to Questions

**Comments to the Author**

1. If the authors have adequately addressed your comments raised in a previous round of review and you feel that this manuscript is now acceptable for publication, you may indicate that here to bypass the “Comments to the Author” section, enter your conflict of interest statement in the “Confidential to Editor” section, and submit your "Accept" recommendation.

Reviewer #1: All comments have been addressed

Reviewer #3: (No Response)

Reviewer #4: (No Response)

2. Is the manuscript technically sound, and do the data support the conclusions?

Reviewer #1: Yes

Reviewer #3: (No Response)

Reviewer #4: Partly

3. Has the statistical analysis been performed appropriately and rigorously? 

Reviewer #1: Yes

Reviewer #3: (No Response)

Reviewer #4: Yes

4. Have the authors made all data underlying the findings in their manuscript fully available?

Reviewer #1: Yes

Reviewer #3: (No Response)

Reviewer #4: Yes

5. Is the manuscript presented in an intelligible fashion and written in standard English?

Reviewer #1: Yes

Reviewer #3: (No Response)

Reviewer #4: No

6. Review Comments to the Author

Reviewer #1: I would like to congratulate the authors for their excellent work in the field of Survey Sampling. They have made a very detailed revision and convinced me with good arguments. Therefore, I recommend this article for the possible publication in PLOSONE.

Reviewer #3: (No Response)

Reviewer #4: You need to get a language and grammar as well as structure editor immediately since the paper is not easy to read. I got lost and was not able to trace back where I had ended.

7. PLOS authors have the option to publish the peer review history of their article (what does this mean?). If published, this will include your full peer review and any attached files.

Reviewer #1: No

Reviewer #3: No

Reviewer #4: No

---

## [Author Response · Author response to Decision Letter 1]

3 Sep 2024

Replies to Referee’s Comments

Double Extreme-Cum-Median Ranked Set Sampling

We are thankful to the Editor and the anonymous Reviewers for the useful comments and the opportunity to improve our paper. We have revised the article by addressing all the suggested points. Here are the point-by-point responses to all the comments. In the revised version of the paper, we highlighted the changes in yellow color.

Reviewer #3: 

1) The authors should give the number of each section. For example, “1. Introduction” instead of “Introduction” for the title of Section.

Reply: Thanks for your suggestion. The given suggestion is already incorporated. 

2) The authors should give the number of each page

Reply: Thanks for your suggestion. Each page number has been assigned. 

3) Page 3, line 8, “see Zamanzade and Mahdizadeh (2020)” should be corrected as “see [21]”. 

Reply: Thanks for your suggestion. The said reference is corrected. 

4) The authors should explain the means of “yi(i)j, µy(i)” in Equations 1 and 2. 

Reply: Thanks for your suggestion. The given terms are properly defined after Equation 1 and 2. 

5) As pointed out by two reviewers, please don’t cite so many references together without discussing them as you have given after Equation 2. Some of them can be deleted.

Reply: Thanks for your suggestion. The discussion of the references has been incorporated. 

6) The symbols under Equations 5 and 6 are not clear. 

Reply: Thanks for your suggestion. The symbols under Equations 5 and 6 are cleared.

7) Why the mean estimators under DEMRSS for even and odd sample sizes are unbiased?

Reply: Thank you for the suggestion. The reference is mentioned after Equation 11. 

8) How can we obtain Equations 12 and 13?

Reply: Thank you for the suggestion. The derivations/proves of Equation 12 and 13 are incorporated in Appendix I.

---

## [Decision Letter · Decision Letter 2]

1 Oct 2024

Double Extreme-Cum-Median Ranked Set Sampling

PONE-D-24-07535R2

Dear Dr. Iqbal,

We’re pleased to inform you that your manuscript has been judged scientifically suitable for publication and will be formally accepted for publication once it meets all outstanding technical requirements.

Kind regards,

Yu Zhou

Academic Editor

PLOS ONE

Additional Editor Comments (optional):

Reviewers' comments:

Reviewer's Responses to Questions

**Comments to the Author**

1. If the authors have adequately addressed your comments raised in a previous round of review and you feel that this manuscript is now acceptable for publication, you may indicate that here to bypass the “Comments to the Author” section, enter your conflict of interest statement in the “Confidential to Editor” section, and submit your "Accept" recommendation.

Reviewer #3: (No Response)

Reviewer #4: All comments have been addressed

2. Is the manuscript technically sound, and do the data support the conclusions?

Reviewer #3: (No Response)

Reviewer #4: Yes

3. Has the statistical analysis been performed appropriately and rigorously? 

Reviewer #3: (No Response)

Reviewer #4: Yes

4. Have the authors made all data underlying the findings in their manuscript fully available?

Reviewer #3: (No Response)

Reviewer #4: Yes

5. Is the manuscript presented in an intelligible fashion and written in standard English?

Reviewer #3: (No Response)

Reviewer #4: Yes

6. Review Comments to the Author

Reviewer #3: The authors have revised their paper according my comments. I think this revised version can be accepted for publication.

Reviewer #4: None.

Kindly invest in having an English editor (who will clean the manuscripts, present it in a consistent manner and work on grammar before you send to a journal).

7. PLOS authors have the option to publish the peer review history of their article (what does this mean?). If published, this will include your full peer review and any attached files.

Reviewer #3: No

Reviewer #4: No

---

## [Editor Report · Acceptance letter]

9 Oct 2024

PONE-D-24-07535R2 

PLOS ONE

Dear Dr. Iqbal, 

I'm pleased to inform you that your manuscript has been deemed suitable for publication in PLOS ONE. Congratulations! Your manuscript is now being handed over to our production team.

Kind regards, 

on behalf of

Dr. Yu Zhou 

Academic Editor

PLOS ONE